# Single molecule imaging reveals a major role for diffusion in the exploration of ciliary space by signaling receptors

Fan Ye[1,2], David K Breslow[1], Elena F Koslover[3], Andrew J Spakowitz[3], W James Nelson[1,2]*, Maxence V Nachury[1]*

[1]Department of Molecular and Cellular Physiology, Stanford University School of Medicine, Stanford, United States; [2]Department of Biology, Stanford University, Stanford, United States; [3]Department of Chemical Engineering, Stanford University, Stanford, United States

**Abstract** The dynamic organization of signaling cascades inside primary cilia is key to signal propagation. Yet little is known about the dynamics of ciliary membrane proteins besides a possible role for motor-driven Intraflagellar Transport (IFT). To characterize these dynamics, we imaged single molecules of Somatostatin Receptor 3 (SSTR3, a GPCR) and Smoothened (Smo, a Hedgehog signal transducer) in the ciliary membrane. While IFT trains moved processively from one end of the cilium to the other, single SSTR3 and Smo underwent mostly diffusive behavior interspersed with short periods of directional movements. Statistical subtraction of instant velocities revealed that SSTR3 and Smo spent less than a third of their time undergoing active transport. Finally, SSTR3 and IFT movements could be uncoupled by perturbing either membrane protein diffusion or active transport. Thus ciliary membrane proteins move predominantly by diffusion, and attachment to IFT trains is transient and stochastic rather than processive or spatially determined.

**\*For correspondence:**
wjnelson@stanford.edu (WJN);
nachury@stanford.edu (MVN)

## Introduction

Membrane proteins explore cellular space by diffusion or motor-driven transport on the cytoskeleton, but the relative contributions of these processes in specific cellular contexts have not been directly assessed. A prototypical system to study how proteins explore cellular space is the primary cilium, a surface-exposed compartment consisting of a microtubule-based axoneme ensheathed within a plasma membrane protrusion (*Satir and Christensen, 2007*). The primary cilium is an important signaling compartment (*Goetz and Anderson, 2010*): entire signaling cascades, from the most upstream membrane-embedded receptors to the ultimate effectors, are dynamically concentrated within the ciliary space (*Corbit et al., 2005*; *Haycraft et al., 2005*; *Rohatgi et al., 2007*), and cilium dysfunction abolishes specific pathways such as Hedgehog signaling (*Huangfu et al., 2003*).

The assembly of cilia, and structurally related flagella, relies on intraflagellar transport (IFT), a process in which moving IFT trains bring cargo, such as tubulin and axonemal precursors, to the tip of the axoneme by kinesin II motor-powered transport along axonemal microtubules (*Cole, 2008*). In reverse, polypeptides move by dynein 1b-powered IFT from the tip to the base of cilia. The requirement of the IFT machinery for ciliary assembly, and the bidirectional movements of IFT trains in mature cilia have been substantiated in every model system tested (*Pedersen and Rosenbaum, 2008*) and it is generally accepted that soluble and membrane-embedded cargo explore the ciliary space by loading onto IFT trains at one end of the primary cilium and unloading at the other end (*Qin et al., 2004*; *Pedersen et al., 2006*). Yet, there is limited direct experimental evidence to support this model of IFT-mediated transport of ciliary membrane proteins.

**eLife digest** Primary cilia are tiny protrusions from the cell surface, which have a central role in processing sensory stimuli, such as light or odorants. Cilia are also involved in mediating the response to developmental signaling molecules, including Sonic Hedgehog, and may help to convert mechanical signals into electrical or chemical ones. Primary cilia are made up of an axoneme—a core structure that consists of microtubules extending along the length of the cilium— ensheathed by a membrane that contains a number of receptor proteins.

These receptor proteins travel up and down the cilium, and it is generally assumed that an active process known as intraflagellar transport is responsible for their movement. This process is mediated by motor proteins called kinesins and dyneins, which carry cargo proteins along axonemal microtubules. However, it has been difficult to study the transport of individual receptor proteins directly because they are uniformly distributed over the membranes of the cilia.

Now, Ye et al. have shown that intraflagellar transport is not the most important mode of transport for membrane proteins within primary cilia. By labelling individual receptors with a fluorescent dye and then filming their movements under a microscope, Ye et al. found that the receptors generally did not show the directed, linear motion that would be expected from intraflagellar transport. Instead, much of their movement occurred through passive diffusion, with occasional short bursts of directed motion.

To investigate how rapidly receptor molecules could move through the cilium in this way, Ye et al. used a technique called fluorescence recovery after photobleaching (FRAP). This involves using light to bleach the fluorescent dye attached to receptor molecules in part of the cilium, and then measuring how long it takes for the fluorescence to return as a result of other labelled molecules moving into the bleached area: the shorter this time, the faster the movement of the molecules. It took less than a minute for fluorescence to be restored within a primary cilium, indicating that passive diffusion with occasional active transport can move proteins rapidly through the structure.

By using drugs to inhibit intraflagellar transport, Ye et al. confirmed that the majority of membrane protein transport within primary cilia occurs via diffusion. Further studies are now required to determine whether this is also the case for other molecules that travel along cilia, and whether intraflagellar transport may have a more important role in the assembly of these structures.

Pioneering studies using the flagellated green alga *Chlamydomonas reinhardtii* showed that externally applied beads, which artificially cluster mating receptors, are transported along the flagellar axoneme (***Bloodgood et al., 1979***; ***Bloodgood and Salomonsky, 1989***), but a role for IFT has not been tested in this context. While bulk imaging of some GFP-tagged ciliary membrane proteins (e.g., PKD2-GFP, OSM9-GFP) discerned a few fluorescence foci moving processively in cilia (***Ou et al., 2005***; ***Huang et al., 2007***), ciliary membrane proteins are generally distributed homogeneously inside cilia and information on their dynamics is therefore lacking. Furthermore, while IFT trains have been visualized as large polymeric assemblages by electron microscopy (***Pigino et al., 2009***) and as clusters of fluorescence by light microscopy (***Pedersen and Rosenbaum, 2008***), the IFT entities that ferry cargoes remain elusive.

The goal of our work was to directly measure the contributions of IFT and diffusion to the movement of individual ciliary membrane proteins. We developed an experimental system to image single membrane receptors together with IFT trains in the primary cilium of live cells. By selectively disrupting membrane protein diffusion or motor-driven active transport, we show that cargo and IFT movements can be uncoupled from each other and that diffusion is sufficient for membrane proteins to explore the ciliary surface.

## Results

### Membrane proteins move in a saltatory manner in cilia

To overcome the limitations of ensemble measurements of cargo proteins in primary cilia, we quantified the ciliary movements of individually labelled membrane proteins. We selected two ciliary membrane proteins: the G protein-coupled receptor (GPCR), Somatostatin receptor 3 (SSTR3) (***Händel et al.,***

*1999*), and the Hedgehog signaling intermediate Smoothened (Smo), a non-GPCR 7-pass membrane protein (*Corbit et al., 2005*). To facilitate cilia visualization, SSTR3 and Smo were tagged at the intracellular C-terminus with a fluorescent protein (SSTR3-GFP and Smo-YFP). In order to detect single molecules we also fused a biotinylation acceptor peptide (AP) to the extracellular N-terminus of SSTR3 or Smo, and co-expressed the biotin ligase BirA in the ER of the stable cell lines (*Howarth and Ting, 2008*). Proteins were expressed stably and at low levels in IMCD3-Flp-In cells.

Singly biotinylated SSTR3 and Smo molecules were labelled on the cell surface by extracellular addition of a low concentration (50 pM) of monovalent streptavidin (mSA; *Howarth et al., 2006*) conjugated to Alexa647 fluorescent dye (mSA-A647) (*Figure 1A*). Under these conditions, SSTR3-GFP and Smo-YFP were distributed homogeneously in cilia, but each cilium contained between zero and five dots of mSA-A647 labelled molecules (referred hereafter as single SSTR3 or Smo molecule) (*Figure 1B*); cilia orientation was determined by co-expressing pericentrin-RFP (PCNT; *Gillingham and Munro, 2000*) which localizes to the basal body at the base of the cilium (*Figure 1B*). Single molecule imaging was performed at 2 Hz until fluorescence of mSA was lost due to bleaching (30–60 s). Time-space plots (kymographs; *Figure 1B*, bottom; *Figure 1–figure supplement 1*) were generated from live cell time-lapse images to describe the movement of single molecules.

To our surprise, single molecules of SSTR3 or Smo moved in a saltatory manner that comprised mostly diffusive behavior interspersed with short periods of directional movements (*Figure 1B*, *Figure 1—figure supplement 1*; *Video 1*, and *Video 2*). By using bright foci of RFP-tagged IFT88 as markers of motor-driven IFT trains, we found that the saltatory movement of SSTR3 was clearly different from the smooth, processive movement of IFT trains in the same cilia (*Figure 1C*). Furthermore, not only were the movements of IFT trains and single SSTR3 molecules different, their tracks rarely overlapped even when single SSTR3 molecules displayed short directional movements (*Figure 1D*, and *Figure 1—figure supplement 2*). These observations indicate that SSTR3 and Smo moved primarily by diffusion in the ciliary membrane, and that very small IFT trains might actively move single SSTR3 molecules; these IFT nanotrains are below the sensitivity limit of our imaging.

## Saltatory dynamics is compatible with rapid exploration of the ciliary space

To determine how rapidly single SSTR3 and Smo molecules explore the entire ciliary space, in spite of their saltatory movements, we assessed their mobility by fluorescence recovery after photobleaching (FRAP). We had previously uncovered a diffusion barrier at the ciliary base by showing that photobleaching Smo-YFP or SSTR3-GFP in the entire cilium led to no detectable recovery from its plasma membrane pool (*Hu et al., 2010*). However, photobleaching Smo-YFP (*Hu et al., 2010*) or SSTR3-GFP (*Figure 1C*, *Figure 1—figure supplement 3*, and *Video 3*) in the distal half of the cilium led to fluorescence recovery in less than 60 s at the expense of signal diffusing from the proximal, unbleached half (*Hu et al., 2010*). While this experiment did not distinguish between diffusion and active transport, it showed that the saltatory movements of SSTR3 and Smo are compatible with rapid exploration of the entire ciliary space.

Half-cilium FRAP allowed us to calculate an apparent diffusion coefficient of 0.25 µm²/s for SSTR3, which reflects a mixture of active transport and diffusion ('Materials and methods'). Importantly, comparing the mobility of SSTR3 in the presence or absence of saturating amounts of mSA showed that our single molecule labeling strategy did not affect SSTR3 mobility (*Figure 2A–C*). In addition, our tagging and labeling strategy does not disrupt the signaling properties of SSTR3 since AP-SSTR3-GFP underwent agonist-induced endocytosis from the plasma membrane, a common feature of GPCRs upon activation (*Figure 2D*). These results lead us to conclude that our single molecule imaging reflects the dynamics of functional SSTR3 molecules.

## Relative contributions of active transport and diffusion to protein movements inside cilia

To determine the proportion of time that SSTR3 and Smo spend undergoing active transport we used a statistical analysis of instant velocities, which overcomes the intrinsic bias in manually selecting processive segments from the single molecules kymographs. The combination of motor-driven velocities and diffusive velocities found in live cells was deconvolved by effectively removing motor-driven transport, thus isolating diffusive events. Since the IFT motors kinesin II and dynein 1b require ATP to power active transport, we depleted ATP using two independent approaches. First, we added antimycin and deoxyglucose to cells to interrupt mitochondrial respiration and anaerobic glycolysis. This treatment

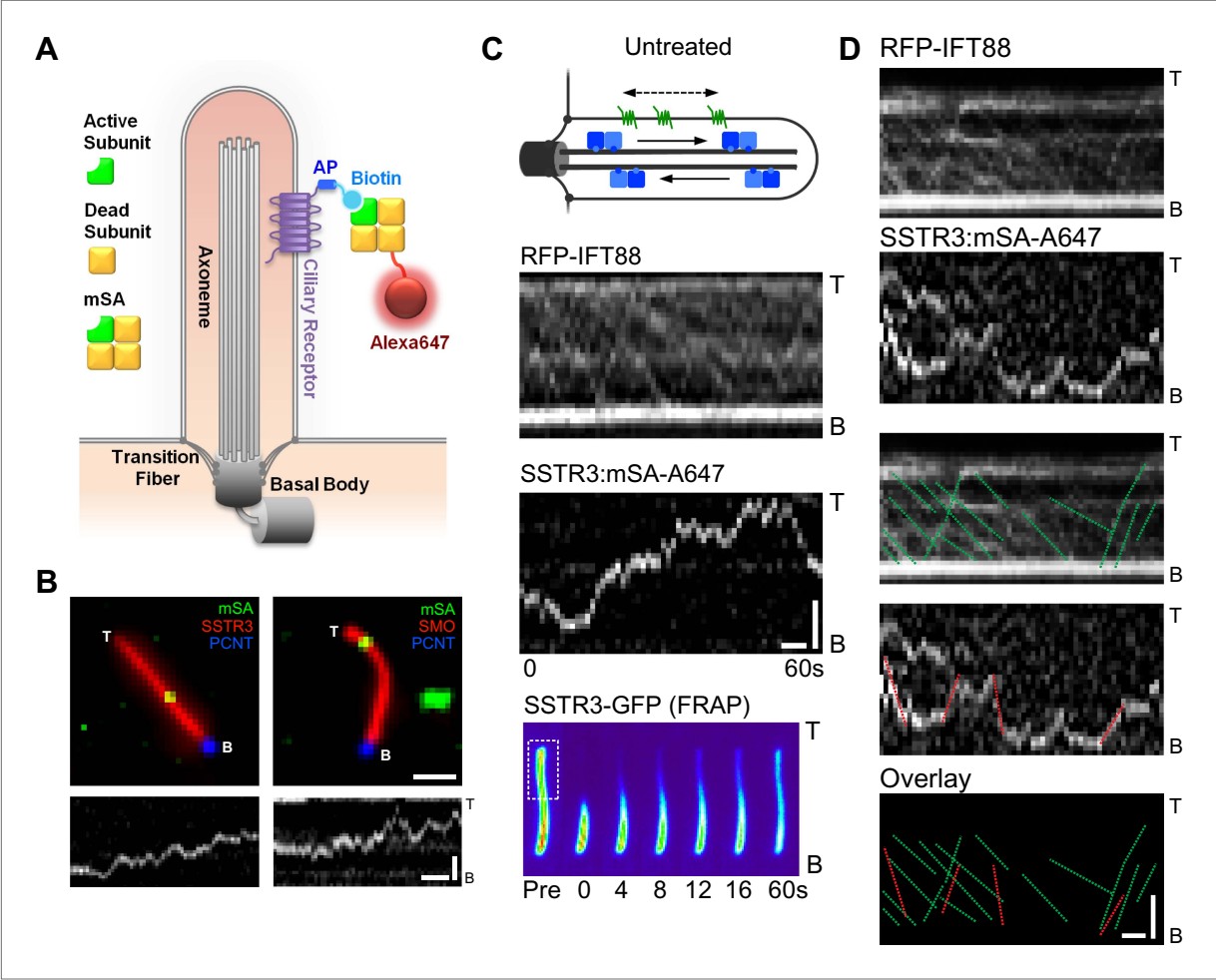

**Figure 1**. Real-time imaging of single signaling receptors in cilia of live cells. (**A**) Schematic of single molecule labeling strategy. SSTR3 or Smo were fused at the extracellular N-terminus to an acceptor peptide (AP) for the biotin ligase BirA. Biotinylated AP-SSTR3 and AP-Smo molecules were sparsely revealed by Alexa647-conjugated monovalent streptavidin (mSA-Alexa647) added at low concentrations (50 pM) to the extracellular medium. (**B**) IMCD3 cells stably expressing AP-SSTR3-GFP (SSTR3, pseudo-colored red, left panel) or AP-Smo-YFP (SMO, pseudo-colored red, right panel) were transfected with Pericentrin-RFP (PCNT, pseudo-colored blue) to mark the ciliary base and BirA to biotinylate AP-SSTR3-GFP. Biotinylated SSTR3 or Smo were detected with mSA-Alexa647 (mSA, pseudo-colored green). The kymograph represents the movement of a single mSA-Alexa647 labeled AP-SSTR3-GFP or AP-Smo-YFP in live cells. The tip (T) and the base (B) of the cilium are indicated. Scale bars, 2 µm (y), 4 s (x). (**C**) Kymographs of simultaneous live cell imaging of TagRFP.T-IFT88 (RFP-IFT88, IFT train) and single molecule SSTR3 (SSTR3:mSA-A647) movement in untreated cells. The mobility of ciliary SSTR3 was assessed by half-cilium FRAP (montage of heat-maps, bottom). Scale bars, 2 µm (y), 5 s (x). (**D**) Comparison of IFT88 foci track with single SSTR3 directional tracks in untreated cells. The processive movement of mSA labeled SSTR3 (SSTR3:mSA-A647, red dashed line) and IFT88 foci tracks (RFP-IFT88, green dashed line) are indicated. Little overlap is observed between IFT88 foci tracks and single SSTR3 tracks in untreated cells.

The following figure supplements are available for figure 1:

**Figure supplement 1**. Additional kymographs.

**Figure supplement 2**. Additional dual channel kymographs.

**Figure supplement 3**. Half-cilium FRAP.

reduced cellular ATP levels by 90% (*Figure 3C*), indicating that some ATP generation system(s) remained (e.g., phosphocreatine). Nevertheless, this treatment inhibited all IFT88 foci movements (*Figure 3A*). However, the saltatory movements of single SSTR3 molecule in the same cilia were unaffected (*Figure 3B*, and *Figure 3—figure supplement 1*).

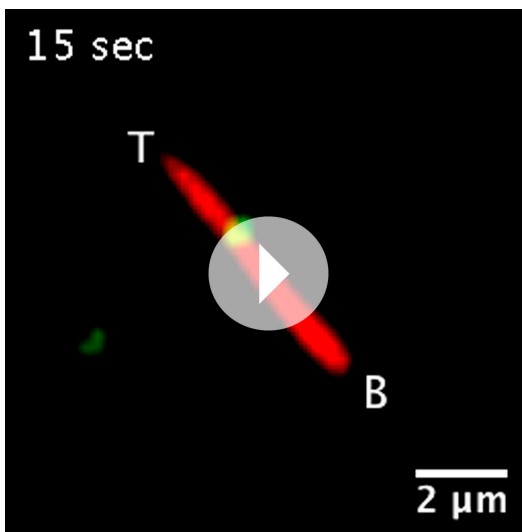

**Video 1**. Live cell imaging of a single ciliary SSTR3 molecule. IMCD3 cells stably expressing AP-SSTR3-GFP (pseudo-colored red) were transfected with BirA-ER to biotinylate AP-SSTR3-GFP. Biotinylated SSTR3 was detected with mSA-Alexa647 (mSA, pseudo-colored green). The tip (T) and the base (B) of the cilium are indicated. Scale bar, 2 μm.

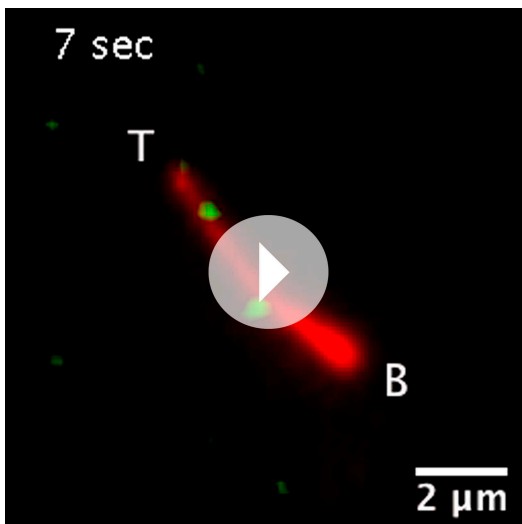

**Video 2**. Live cell imaging of a single ciliary Smo molecule. IMCD3 cells stably expressing AP-Smo-YFP (Smo, pseudo-colored red) were transfected with BirA-ER to biotinylate AP-Smo-YFP. Biotinylated Smo was detected with mSA-Alexa647 (mSA, pseudo-colored green). The tip (T) and the base (B) of the cilium are indicated. Scale bar, 2 μm.

To reveal the fraction of SSTR3 molecules undergoing active transport, we conducted a differential analysis of single SSTR3 molecule movements in cilia of control vs antimycin/deoxyglucose-treated cells. Since kinesin II and dynein 1b each move at characteristic velocities (**Scholey, 2008**), motor-driven movements of single SSTR3 molecules should be enriched in the distribution of instant velocities along the cilium in control cells vs antimycin/deoxyglucose-treated cells (**Figure 3D**). Differences between distributions were analyzed by performing an effective histogram subtraction ('Materials and methods'). Anterograde and retrograde velocities were binned separately to create histograms with an optimum bin size that generated an unbiased estimate of the distributions (**Scott, 1979**). Using nonlinear least-squares regression, the live cell velocity histogram was fitted to a mixed distribution consisting of fraction f of a Gaussian with unknown mean and variance and fraction $(1-f)$ of the antimycin/deoxyglucose-treated cell distribution. This, and two other ('Materials and methods'), unbiased statistical tests were highly significant for both anterograde and retrograde SSTR3 velocities, and indicated that 27% of anterograde and 13% of retrograde movements of SSTR3 involved active transport (**Figure 3D**).

Second, we used the cholesterol-selective detergent digitonin. which selectively permeabilizes the plasma membrane but leaves the ciliary membrane intact (**Breslow et al.**). This results in depletion of cytoplasmic ciliary contents including motors and ATP (**Figure 4A**, top panel). As expected, this treatment reduced the ATP concentration to undetectable levels (**Figure 4E**, and **Figure 4—figure supplement 1**). Significantly, IFT foci movements were completely inhibited, while SSTR3 molecules retained their saltatory behavior (**Figure 4A**, and **Figure 4—figure supplement 2A**). Again, three unbiased statistical tests found highly significant differences in the distribution of instant velocities between untreated and digitonin-treated cells. Applying the same effective subtraction of instant velocities as done for antimycin/deoxyglucose-treated cells, we found that 22% of anterograde and 24% of retrograde movements of SSTR3 involved active transport (**Figure 4B**). Importantly, the mean of the Gaussians representing the differences of instant velocities between untreated and digitonin-permeabilized cells aligned closely with the mean velocities of IFT foci movements (**Figure 4D**). This strongly suggests that the active transport events detected by our statistical subtraction are powered by the IFT motors kinesin II and dynein 1b.

**Video 3**. Representative examples of half-cilium FRAP in different conditions. The distal half of the cilia were photobleached and the recovery rate of the SSTR3-GFP fluorescent signal in the bleached region was recorded at 1 s interval. Scale bar, 2 µm.

While the proportions of active anterograde transport events uncovered by digitonin permeabilization and antimycin/deoxygluocose treatments are very similar, the proportions of active retrograde movements identified by the two treatments were different. A possible interpretation is that dynein 1b retains some ability to operate under load at low ATP levels while kinesin II is fully inactive when ATP levels are reduced 10-fold. Interestingly, FRAP analysis showed that the mobility of SSTR3-GFP was similar in untreated and digitonin-treated cells (*Figure 4A*, *Figure 4—figure supplement 3*, and *Video 3*). This result strongly suggests that the combination of passive diffusion and active transport (by nanotrains, see above) are effective at enabling membrane proteins to explore the ciliary space.

To assess whether the proportion of time spent by SSTR3 in active transport vs passive diffusion was similar for other proteins, we applied our permeabilization strategy to AP-Smo-YFP expressing cells (*Figure 4—figure supplement 2B*). Again, the three statistical tests all showed a significant difference in the distribution of instant velocities between untreated and digitonin-treated cells and the Gaussians

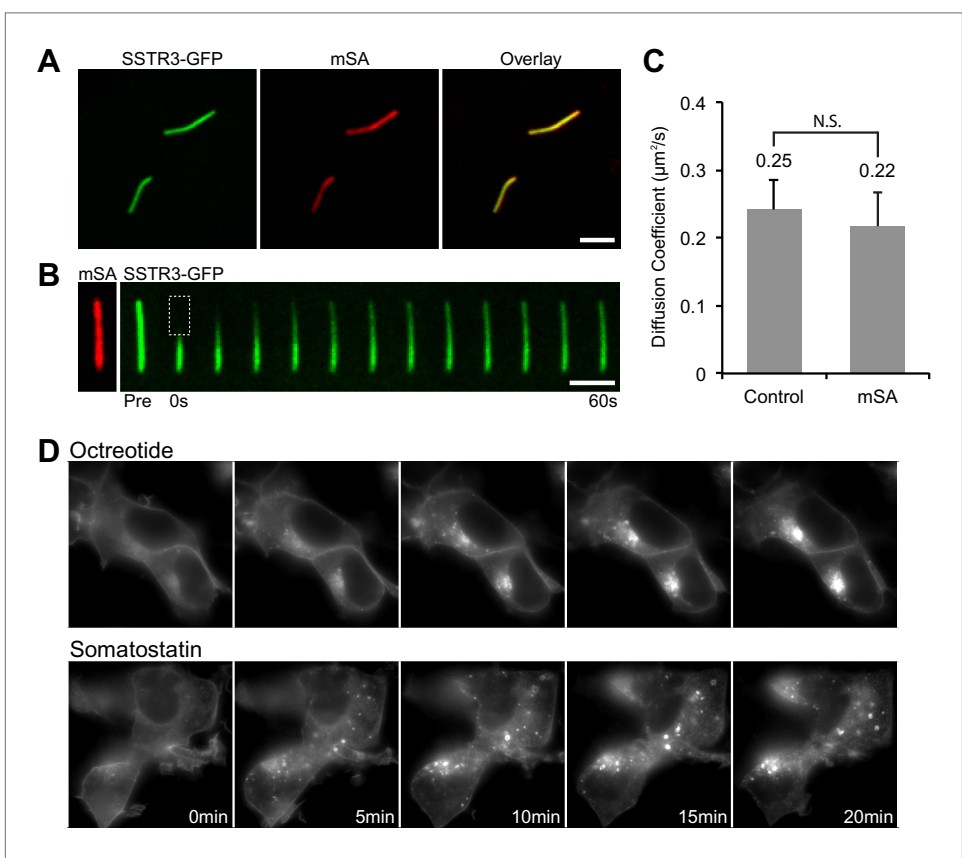

**Figure 2**. Functionality of the AP-SSTR3-GFP fusion protein. (**A**) Saturated labeling of biotinylated AP-SSTR3-GFP (SSTR3-GFP) with 20 nM of mSA-Alexa647 (mSA) for 1 hr. Scale bar, 2 µm. (**B**) Time series montage represents the mobility of ciliary AP-SSTR3-GFP saturatedly labeled with mSA-Alexa647 after photobleaching. Scale bar, 2 µm. (**C**) The diffusion coefficients quantified from control cells vs cells with saturated staining of mSA-Alexa647 (mSA). More than 10 cilia were analyzed for each condition. Error bars, SD. p>0.05. (**D**) Live cell imaging of HEK293T cells expressing AP-SSTR3-GFP. Cells were treated with 10 µM Octreotide or Somatostatin immediately before imaging.

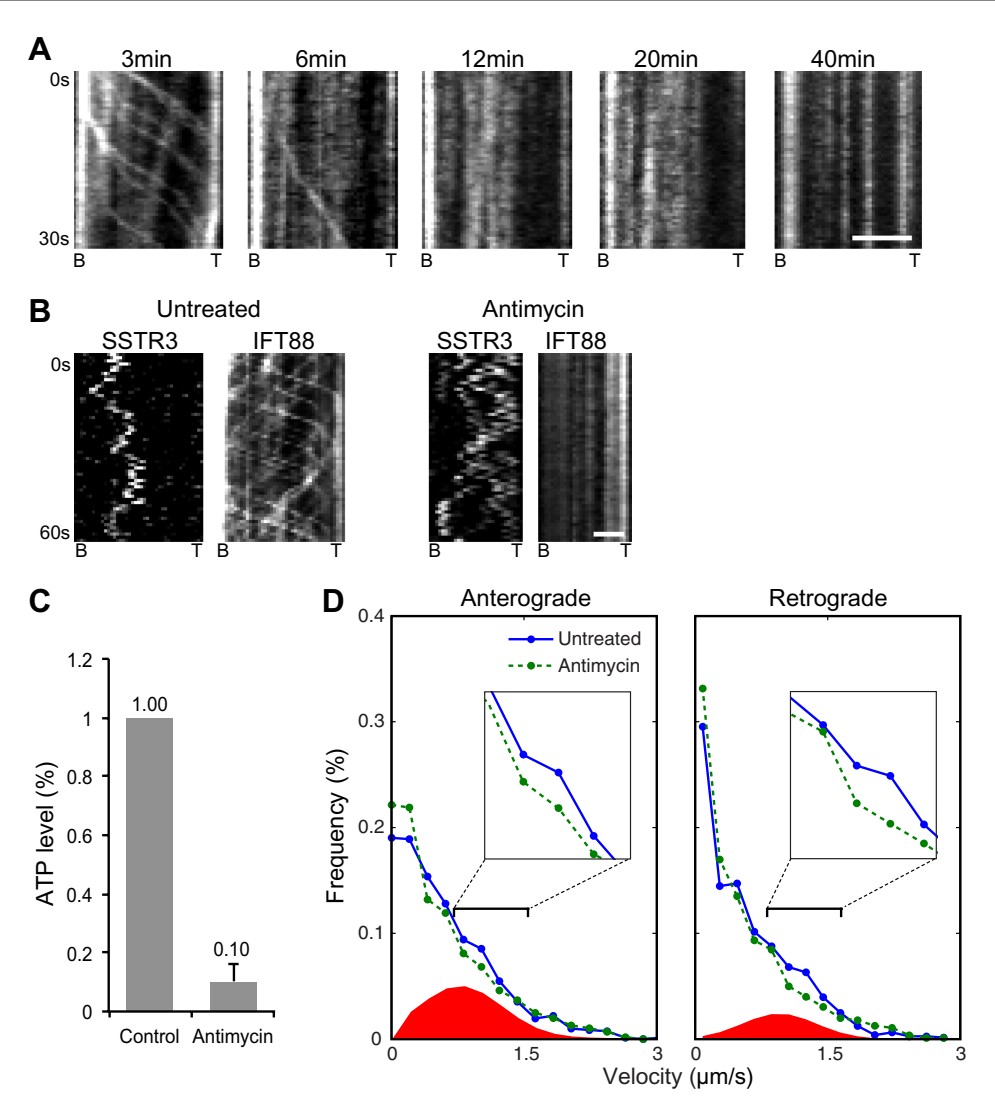

**Figure 3**. Single molecule imaging in ATP-depleted cells reveals the receptor population undergoing active transport in live cells. (**A**) Kymographs of GFP-IFT88 foci movements after Antimycin A and 2-deoxyglucose (2DG) treatment. Scale bar, 2 μm. (**B**) Kymographs of simultaneous live cell imaging of tagRFP.T-IFT88 (IFT88) and single molecule SSTR3 (SSTR3) movement before and after 40 min of Antimycin A and 2DG treatment. Scale bar, 2 μm. (**C**) ATP levels quantified by luciferin-luciferase bioluminescence assay normalized to the levels in control-treated cells. To deplete intracellular ATP, IMCD3 cells were treated with 20 μM Antimycin + 10 mM 2DG for 40 min. Each treatment was measured in triplicate. (**D**) Instant velocity distributions of single SSTR3 movements in untreated vs ATP-depleted cells. Statistical analyses show a significant difference between velocity distributions of untreated and ATP-depleted cells for both the anterograde velocities (p=0.03) and retrograde velocities (p=0.03). The live cell data (blue) was fitted to a mixed model combining the ATP-depleted data (green) and an additional Gaussian distribution (red), with the latter found to contribute a fraction of 26.6 +/− 5.9% (anterograde) and 12.8 +/− 3.1% (retrograde). n>1200.

The following figure supplements are available for figure 3:

**Figure supplement 1**. Additional kymographs.

representing active transport amounted to 34% (anterograde) and 32% (retrograde) of the entire distributions (***Figure 4C***). While the proportions of time undergoing active transport are slightly greater for Smo than for SSTR3, they indicate that Smo also spends the majority of its time diffusing in the ciliary membrane rather than undergoing directed IFT.

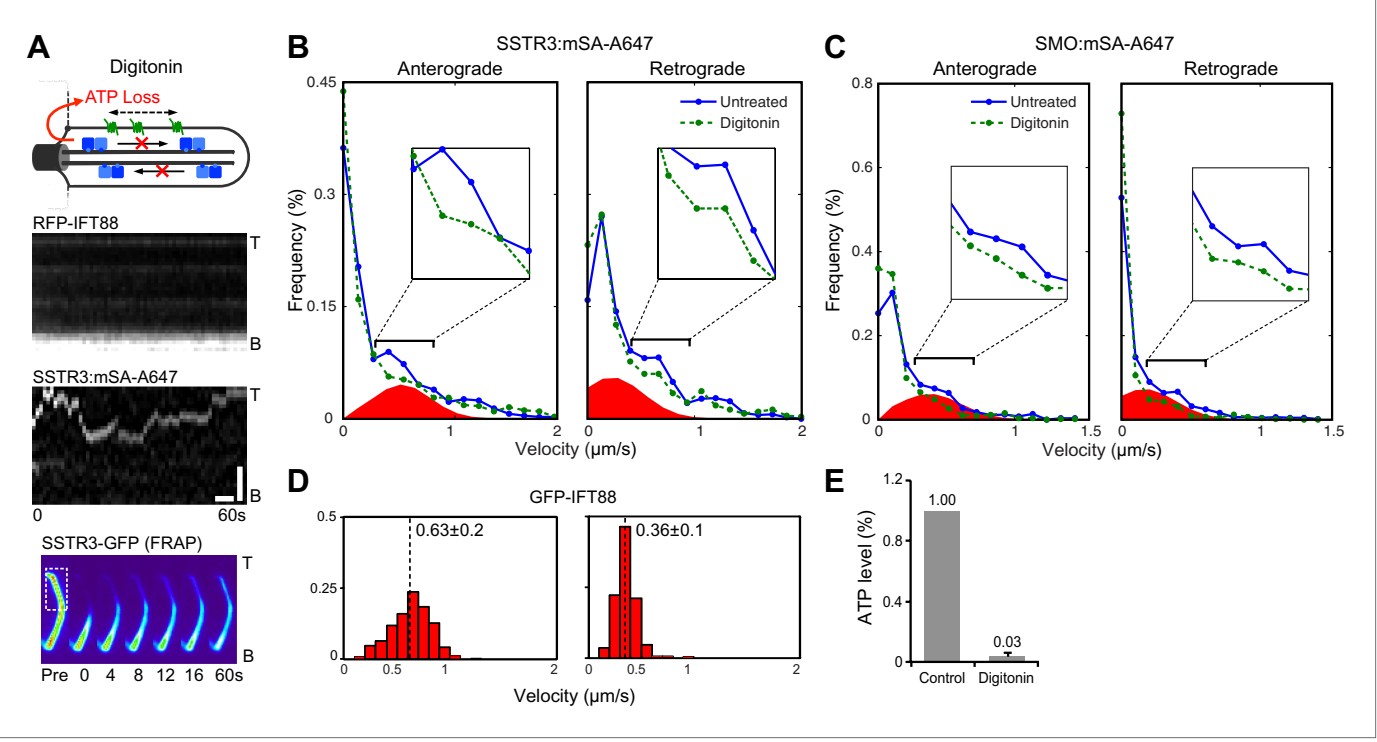

**Figure 4**. Single molecule imaging in digitonin-permeabilized cells reveals the receptor population undergoing active transport in live cells. (**A**) Kymographs of simultaneous live cell imaging of TagRFP.T-IFT88 (RFP-IFT88, IFT trains) and single molecule SSTR3 (SSTR3:mSA-A647) movements in digitonin semi-permeabilized cells. The immobilization of ciliary SSTR3 was confirmed by half-cilium FRAP (montage of heat-maps, bottom). Scale bars, 2 μm (y), 5 s (x). (**B**) Instant velocity distribution of single SSTR3 movements along cilia in untreated and digitonin semi-permeabilized cells. The live cell data (blue) was fitted to a mixed model combining the permeabilized data (green) and an additional Gaussian distribution (red), with the latter found to contribute a fraction of 21.8 +/− 12.6% (anterograde) and 24.3 +/− 5.5% (retrograde). n>1200. (**C**) Instant velocity distribution of single Smo movements along cilia in untreated and digitonin semi-permeabilized cells. The live cell data (blue) was fitted to a mixed model combining the permeabilized data (green) and an additional Gaussian distribution (red), with the latter found to contribute a fraction of 32.1 +/− 5.9% (anterograde) and 34.1 +/− 1.9% (retrograde). n>1200. (**D**) Velocity distribution of GFP-IFT88 foci movements measured from kymographs. The mean velocities are shown in the plot. n>290. (**E**) ATP levels quantified by luciferin-luciferase bioluminescence assay normalized to the levels in control-treated cells. To deplete intracellular ATP, IMCD3 cells were permeabilized with 60 μg/ml digitonin for 7 min. Each treatment was measured in triplicate.

The following figure supplements are available for figure 4:

**Figure supplement 1**. Measurement of ATP levels.

**Figure supplement 2**. Additional kymographs.

**Figure supplement 3**. Digitonin permeabilization does not affect the mobility of SSTR3 in cilia.

## Ciliobrevin inhibits IFT foci movements without perturbing active transport of SSTR3

An alternative strategy to inhibit active transport is to use ciliobrevin (*Figure 5A*), which selectively inhibits dynein but not kinesins (*Firestone et al., 2012*). Treatment of cells with ciliobrevin resulted in the rapid and progressive inhibition of retrograde movement of IFT88 foci (within approximately 2–3 min; *Figure 5B*), and after 30 min anterograde movements were also inhibited (*Figure 5A*). Inhibition of anterograde IFT88 movement by ciliobrevin was unexpected since this direction of movement involves kinesin II, which is not targeted by ciliobrevin (*Firestone et al., 2012*). It is possible that dynein is involved in the delivery of IFT complexes (including kinesin II) from the cytoplasm to the base of the cilium for anterograde transport, and that inhibition of dynein gradually reduces the replenishment of these complexes in the cilium. Ciliobrevin did not affect single SSTR3 molecule movements in the

ciliary membrane (*Figure 5A*, *Figure 5—figure supplement 1* and *Video 3*); moreover, the same statistical tests that had been performed on the data from the digitonin and the antimycin/deoxyglucose treatments (*Figure 3D*, and *Figure 4B,C*) failed to uncover a significant difference between the distributions of SSTR3 velocities in control and ciliobrevin-treated cells (*Figure 5C*). FRAP analysis also showed that the mobility of SSTR3-GFP (*Figure 5A*, *Figure 5—figure supplement 2*, and *Video 3*) was similar in untreated and ciliobrevin-treated cells.

Given prior models of IFT-mediated protein transport (*Ou et al., 2005*; *Huang et al., 2007*), it is surprising that ciliobrevin inhibited motor-driven movements of fluorescent IFT88 foci, but not those of single SSTR3 molecules. This raises questions about the relationship between IFT and active transport of SSTR3, especially since directional tracks of single SSTR3 molecules rarely overlapped with movements of large IFT88 foci in control cells (*Figure 1D*, and *Figure 1—figure supplement 2*). As noted above, it is possible that the majority of directional SSTR3 movements are mediated by IFT nanotrains that are smaller than the IFT88 foci that we can resolve, and which might be less sensitive to dynein inhibition. Although ciliobrevin has been shown to inhibit dynein in gliding assays (*Firestone et al., 2012*), it is also possible that it is less potent against dynein under specific loads such as those corresponding to very small IFT nanotrains. Another possibility is that directional SSTR3 movements occurred on other motors (*Hao et al., 2011*) that are not inhibited by ciliobrevin.

## Immobilizing ciliary membrane proteins does not immobilize IFT trains

Having found that IFT inhibition by three independent methods failed to affect SSTR3 movements, we hypothesized that the coupling between single SSTR3 molecules and IFT trains is weak and transient. This predicts that the movement of IFT trains is a constitutive process largely independent of binding to cargo such as SSTR3. Since surface-exposed membrane proteins like SSTR3 are glycosylated, multivalent lectins such as wheat germ agglutinin (WGA) will cross-link and immobilize them in the ciliary membrane (*Golan et al., 1986*). Indeed, addition of WGA completely stopped SSTR3 movements within minutes as measured by single molecule imaging and half-cilium photobleaching (*Figure 6A*, *Figure 6—figure supplements 1 and 2*, and *Video 3*). However, WGA addition had only a small, albeit statistically significant effect on IFT movements (*Figure 6B–D*, and *Video 4*). Since WGA binds to nearly all membrane proteins, we can extrapolate these results to conclude that IFT88 movements can be largely uncoupled from binding to ciliary membrane protein cargo.

## Discussion

By following the movements of single membrane proteins in the primary cilium, we find that previous models of cargo loading onto IFT trains are no longer tenable. While every evidence point to IFT trains assembling at one end of the cilium and then being transported by an IFT motor all the way to the other extremity of the cilium, keeping their state of assembly intact along the way, we find instead that cargo are highly labile components of IFT trains. Our finding that immobilization of membrane proteins (cargoes) only minimally affected IFT train formation and movement suggests that IFT trains assemble spontaneously rather than 'on demand' by binding cargo; this is similar in principle to the cargo-independent assembly of clathrin-coated pits (*Cocucci et al., 2012*). In our model (*Figure 6E*), IFT trains assemble spontaneously at a given frequency at the base and tip of cilia, forming conveyor belts along the length of the cilium onto which membrane cargo can hop on and off. While this model would at first glance appear to contradict prior findings of PKD2-GFP and OSM9-GFP foci moving processively along cilia (*Ou et al., 2005*; *Huang et al., 2007*), it is likely that this is due to differences between ensemble and single molecule imaging. For example, it is possible that each IFT train carries a constant number of cargo molecules but individual cargo molecules continually hop on and off; ensemble imagining would not detect these individual events which we could detect here at the single molecule level. Thus membrane protein transport in cilia may be reminiscent of the slow axonal transport pathway in which most proteins synthesized in the neuronal body move at an apparent slow rate along the axon because they infrequently hop onto motors (*Scott et al., 2011*).

Another interpretation of our data is that proteins undergo advective movements created by the constant movements of large IFT trains up or down the cilium. The influence of fluid motion to particle movements can be measured by the Péclet number $Pe = UR/D$ where $U$ is the speed of fluid motion, $R$ is the radius of the structure and $D$ the diffusion coefficient. If $Pe < 1$, diffusion outpaces the speeds of fluid motion and the contribution of fluid motion to particle transport are minimal (*Goldstein et al., 2008*). In the case of the cilium, $U$ is of the order of IFT motor speeds ($10^{-4}$ cm/s), $R$ is ~$10^{-5}$ cm and $D$

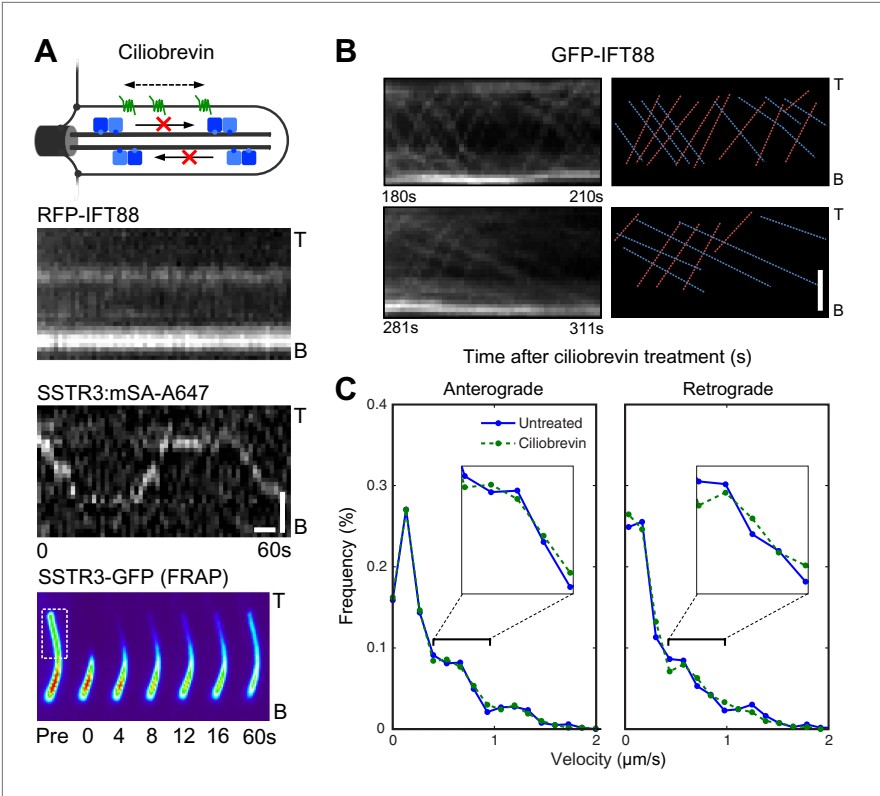

**Figure 5**. Ciliobrevin treatment abolishes IFT train movement but not active transport of SSTR3. (**A**) Kymographs of simultaneous live cell imaging of TagRFP.T-IFT88 (RFP-IFT88, IFT complex) and single molecule SSTR3 (SSTR3:mSA-A647) movements in ciliobrevin treated cells (>30 min). The mobility of ciliary SSTR3 was confirmed by half-cilium FRAP (montage of heat-maps, bottom). Scale bars, 2 μm (y), 5 s (x). (**B**) Early time course of GFP-IFT88 foci movements after ciliobrevin D treatment. The velocity of retrograde tracks (blue dashed line) progressively decreased after 3.5 min, while anterograde foci (red dashed line) movements were unaffected until 5 min and then progressively reduced. (**C**) Instant velocity distributions of single SSTR3 movements in untreated vs ciliobrevin-treated cells. Statistical analyses show no significant difference (p>0.9) between the distributions of velocities of untreated and ciliobrevin-treated cells. n>1200.

The following figure supplements are available for figure 5:

**Figure supplement 1**. Additional kymographs.

**Figure supplement 2**. Ciliobrevin treatment does not affect the mobility of SSTR3 in cilia.

---

is $2 \times 10^{-9}$ cm²/s for SSTR3 and $10 \times 10^{-9}$ cm²/s for a typical protein such as tubulin. The calculated values of Pe are thus 0.5 for SSTR3 and 0.1 for tubulin, thereby suggesting that passive advection plays a very minor role in the movement of typical proteins inside cilia. Moreover, the passive advection model is incompatible with our finding that IFT foci movements can be interrupted without affecting active transport of SSTR3 (*Figure 5*). This experiment suggests that unresolvable IFT nanotrains transport SSTR3 when cells are exposed to ciliobrevin, and that large IFT trains visualized as readily detectable foci of fluorescence may be devoid of cargo and represent recycling IFT complexes. The hypothesis that IFT nanotrains, possibly reduced to single IFT complexes bridging a cargo to a motor, represent the transporting entities is in line with nearly all other example of motor-cargo adaptors (*Hirokawa et al., 2009*). An alternative hypothesis is that ciliobrevin creates an artificial situation in cilia that forces cargo to utilize IFT nanotrains that are not normally found in cilia. A resolution of these possibilities will require advanced imaging approaches that are capable of detecting single molecules of IFT complexes.

In summary, our studies indicate that ciliary membrane proteins such as SSTR3 and Smo explore the ciliary space by passive diffusion rather than only by motor-driven, directional transport. Indeed, the diffusion coefficient of SSTR3 (D= 0.25 μm²/s; *Figure 5—figure supplement 2B*) indicates that a

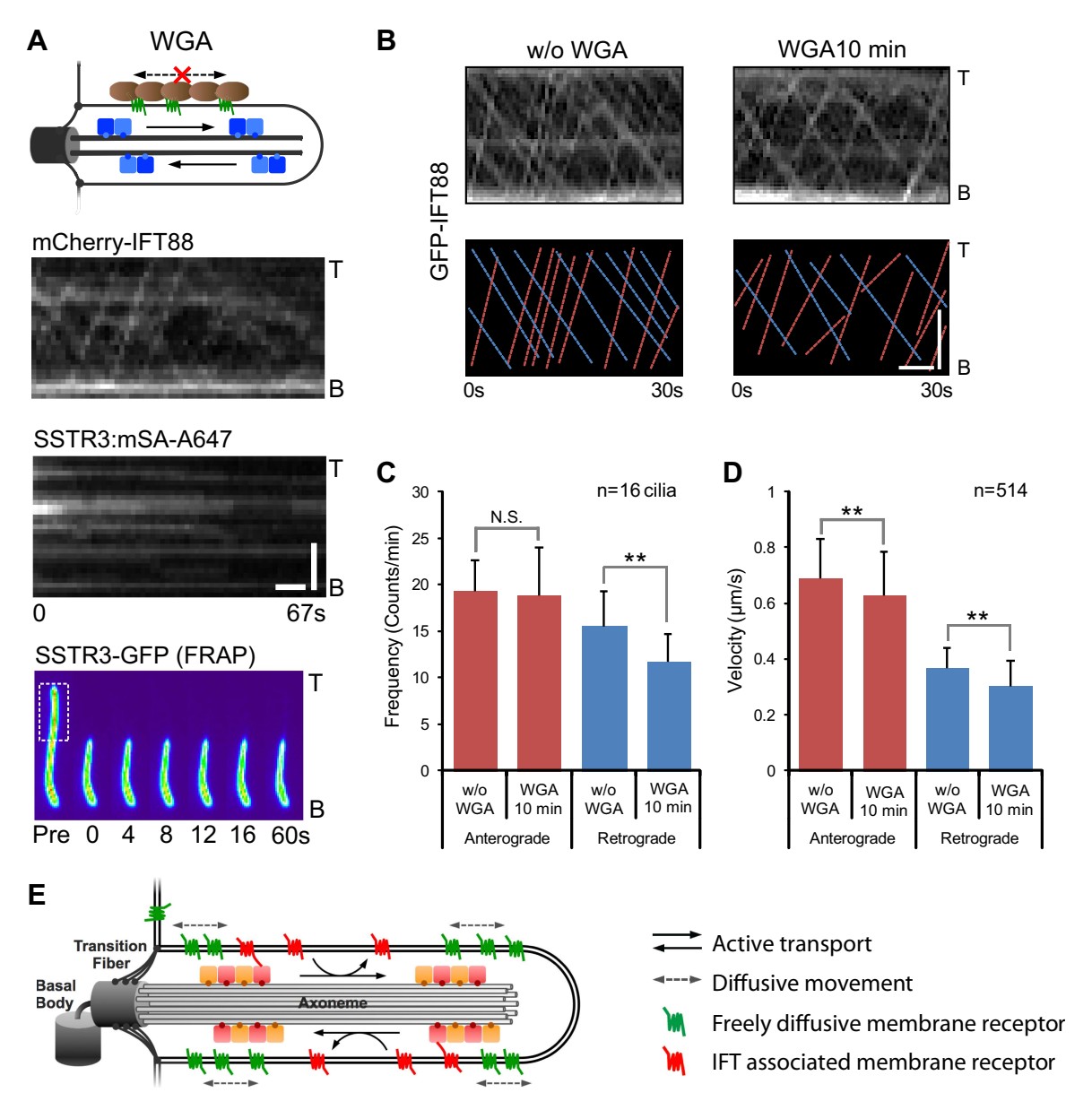

**Figure 6**. Effect of immobilizing membrane proteins on IFT train movements. (**A**) Kymographs of simultaneous live cell imaging of mCherry-IFT88 (IFT complex) and single molecule SSTR3 (SSTR3:mSA-A647) movements in WGA treated cells (>7 min). The immobilization of ciliary SSTR3 was confirmed by half-cilium FRAP (montage of heat-maps, bottom). Scale bars, 2 µm (y), 5 s (x). (**B**) Representative kymographs of GFP-IFT88 fluorescent foci movements before WGA treatment (left, w/o WGA) and after 10 min of WGA treatment (right, WGA 10 min). The anterograde (red dashed lines) and the retrograde (blue dashed lines) movements are indicated in the bottom panels. Scale bars, 2 µm. (**C** and **D**) Bar charts representing the frequency (**C**) and velocity (**D**) of GFP-IFT88 fluorescent foci movements before and after 10 min of WGA treatment in IMCD3 cells. While the differences between WGA-treated and control cells are relatively small, with the exception of the frequency of anterograde trains they are statistically significant; N.S.: $p>0.05$; **$p<0.05$. (**E**) Schematics of ciliary membrane protein dynamics. Single molecule imaging reveals that the majority of ciliary SSTR3 (green) undergo free diffusion, which allows them to explore the ciliary surface efficiently. Only a small portion of ciliary SSTR3 (red) movement is related to IFT. The interaction between SSTR3 and IFT appears to be transient and dynamic.

The following figure supplements are available for figure 6:

**Figure supplement 1**. Additional kymographs.

**Figure supplement 2**. WGA treatment stops the mobility of SSTR3 in cilia.

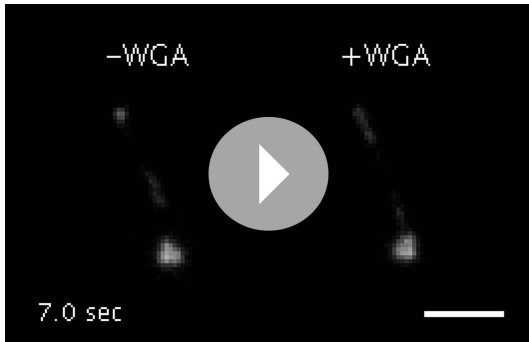

**Video 4**. Live cell imaging of GFP-IFT88 fluorescent foci movement before (−WGA) and after 10 min of WGA treatment (+WGA) in IMCD3 cells. Scale bar, 2 µm.

single SSTR3 molecule could diffuse along the length of a 4 µm cilium in less than 40 s, while IFT-mediated transport would complete the same trip in 8 s. Interestingly, the dimensions of a cilium are similar to those of a bacterium in which proteins rapidly explore the cellular space by diffusion in the absence of cytoskeletal motors (*Elowitz et al., 1999*). The importance of diffusion in exploration of the ciliary space does not rule out other roles for IFT in ciliogenesis, or the possible formation of IFT scaffolds for signaling cascades (*Pan and Snell, 2002*; *Wang et al., 2006*). Further studies of other single molecule dynamics under signaling conditions, using techniques introduced here, will be needed to address these possibilities.

## Materials and methods

### Constructs

Mouse SSTR3-GFP (*Berbari et al., 2008*) (a gift from Dr Kirk Mykytyn, Ohio State University) was fused at its N-terminal extracellular domain to a 13 amino acid acceptor peptide (AP) for the biotin ligase BirA to create AP-SSTR3-GFP. AP-Smoothened-YFP was provided by Carolyn Ott and and Jennifer Lippincott-Schwartz (NIH). AP-SSTR3-GFP or AP-Smoothened-YFP was inserted into pEF5B/FRT-DEST to establish stable expression in FlpIn cells. pDisplay-BirA-ER (*Howarth and Ting, 2008*) and the streptavidin constructs, pET21a-Streptavidin-Alive and pET21a-Streptavidin-Dead (*Howarth et al., 2006*) were provided by Dr Alice Ting (MIT) through Addgene. RFP-pericentrin PACT domain (*Gillingham and Munro, 2000*) was a gift from Dr Sean Munro (MRC Laboratory of Molecular Biology, Cambridge).

### Cell culture, transfection and treatments

Mouse inner medullary collecting duct (mIMCD3) cells were maintained in Dulbecco's modified Eagle medium/F12 (DMEM/F12; Gibco, Grand Island, NY) supplemented with 10% fetal bovine serum (FBS) and 2 mM L-Glutamine at 37°C in 5% $CO_2$. To induce ciliogenesis, cells were cultured in DMEM/F12 supplemented with 0.2% FBS for 24 hr. Lipofectamine 2000 (Invitrogen) was used to transfect plasmids into IMCD3 cells according to the manufacturer's protocol. To block IFT, cells were either treated with 50 µM ciliobrevin D (a gift from Dr James Chen, Stanford University) for 30 min, or semi-permeabilized with highly purified digitonin purchased from EMD Millipore (Billerica, MA) (#300410). Serum-starved IMCD3 cells on coverslips were first placed on an ice-chilled metal block and washed twice with cold Digitonin Assay Buffer (20 mM Hepes, pH 7.4, 115 mM KOAc, 1 mM $MgCl_2$, 1 mM EGTA). For cell permeabilization, coverslips were incubated for 7 min in cold Digitonin Assay Buffer supplemented with 30 µg/ml digitonin and protease inhibitors (10 µg/ml each of leupeptin, pepstatin A, bestatin, aprotinin, AEBSF, and E-64). After permeabilization, coverslips were washed twice with Digitonin Assay Buffer and processed for subsequent imaging assays. For WGA treatment, cells were incubated with 75 µg/ml WGA for 7 min immediately prior to imaging. To test the signaling response of AP-SSTR3-GFP, HEK293T cells were transfected with AP-SSTR3-GFP and maintained in DMEM supplemented with 10% FBS for 24 hr and them treated with 10 µM Octreotide or Somatostatin (SST) immediately before live cell imaging.

### ATP depletion and ATP level measurement

The level of intracellular ATP was determined by luciferin-luciferase bioluminescence assay following instructions from ATP determination kit (A22066; Molecular Probes) with homemade buffers. IMCD3 cells stably expressing AP-SSTR3-GFP were seeded on 35-mm dishes. To deplete intracellular ATP, cells were either incubated in PBS with 20 µM Antimycin A and 10 mM 2-deoxyglucose (2DG) for 40 min or permeabilized with two different concentration of digitonin (30 µg/ml and 60 µg/ml). After treatments, cells were washed twice with PBS (for Antimycin-2DG treatment) or Digitonin Assay Buffer (for digitonin treatment) and placed in ice-cold ATP buffer (20 mM Tris, PH7.5, 0.5% Nonidet P-40, 25 mM NaCl, and 2.5 mM EDTA) for 5 min. Cell lysates were then collected and centrifuged at 13,000×*g* for

15 min at 4°C and protein concentration was measured using Bradford Protein Assay reagent (Bio-Rad, Hercules, CA). For each reaction, 0.5 µg protein was used to measure the ATP level.

### Stable cell lines

An IMCD3 cell line stably expressing AP-SSTR3-GFP or AP-Smo-YFP was generated using the FlpIn system (Life Technologies, Grand Island, NY) as described previously (*Jin et al., 2010*). An IMCD3 host cell line containing a single flippase (Flp) recombination target (FRT) site was transfected with pEF5B/FRT-AP-SSTR3-GFP and the Flp recombinase expression plasmid, pOG44. 48 hr after transfection, cells were selected in DMEM/F12 supplemented with 5 µg/ml blasticidin. Single cell clones were isolated and the expression level of AP-SSTR3-GFP or AP-Smo-YFP in cilia was assessed by fluorescence microscopy. We note that AP-Smo-YFP is constitutively localized to cilia in our cell line because it is expressed at a slightly higher level than the endogenous protein.

### Single molecule labeling of ciliary SSTR3 and Smo

Monovalent streptavidin was purified and conjugated to Alexa647 as described previously (*Howarth et al., 2006*). To singly biotinylate SSTR3 or Smo, IMCD3 cells stably expressing AP-SSTR3-GFP or AP-Smo-YFP were transfected with pDisplay/BirA-ER (BirA-ER is a biotin ligase targeted to the ER lumen) and RFP-Pericentrin PACT domain (a marker of the ciliary base), and maintained in DMEM/F12 supplemented with 0.2% FBS and 10 µM biotin (#BIO200; Avidity) for 24 hr. Excess biotin was removed from cells by washing with PBS. Alternatively, cell surface SSTR3 or Smo can be biotinylated by incubation with 0.3 µM recombinant BirA ligase, 10 µM biotin and 1 mM ATP as described previously (*Howarth and Ting, 2008*). Cells were then incubated with 50 pM mSA-Alexa647 in phenol red-free DMEM/F12 supplemented with 0.2% FBS and 25 mM HEPES (imaging medium) on a shaker at 20 rpm at room temperature for 1 hr. Next, excess mSA-Alexa647 was removed by washing cells with PBS, and cells were then incubated in imaging medium.

### Live cell imaging and FRAP

Cells were seeded on 25 mm diameter cover-glass (Electron Microscopy Sciences, Hatfield, PA) 24 hr before transfection. After transfection, cells were serum-starved for 24 hr, and then the medium was replaced with imaging medium and cells were observed with a DeltaVision system (Applied Precision, Issaquah, WA). The DeltaVision system was equipped with a PlanApo 60×/1.40 objective lens (Olympus, Central Valley, PA), a CoolSNAP HQ2 camera (Photometrics, Tucson, AZ) and an EMCCD camera. To image mSA-Alexa647 labeled SSTR3-GFP, excitation light from the solid state illumination module (InsightSSI) was reduced to 10% intensity with a neutral density filter and the Cy5 channel was exposed for 0.4 s. Half-cilium FRAP was performed as described previously (*Hu et al., 2010*). The GFP fluorescent signal from part of the primary cilium was photobleached with the 488 nm laser from Quantifiable Laser Module (QLM), DeltaVision. The recovery rate of the GFP fluorescent signal in the bleached region was recorded at 1 s interval.

### Data analysis

The mSA-Alexa647 labeled single molecule SSTR3 and Smo were tracked using the SpotTracker plugin in ImageJ (*Sage et al., 2005*). The ciliary base (labeled by Pericentrin-RFP) was used as a reference point to distinguish between anterograde (particle moving away from the base) and retrograde movement (particle moving toward the base).

#### Distribution of instantaneous velocities

For each protein trajectory within a single cilium, principal component analysis was performed on the set of observed positions as well as the position of the cilium base. The largest principal component was used to define the axis of the cilium. Those trajectories that were essentially stationary, that is, with a variance <0.1 $\mu m^2$ along the cilium axis, were designated as outliers and were not included in the analysis (a total of 4 trajectories from permeabilized samples).

Instantaneous velocities were identified as differences between consecutive positions, divided by the time-step of 0.46 s. Each velocity vector was then divided into components lying along and perpendicular to the cilium axis. The distribution of velocities was compared between untreated cells and either digitonin semi-permeabilized cells or ciliobrevin D-treated cells. A two-sample Kolmogorov-Smirnov statistical test was used to determine whether there was a significant difference between these distributions. This test is non-parametric, makes no assumptions about the shape of the

underlying distributions, and does not require binning of the data. For velocity components perpendicular to the cilia, no significant difference was found (p>0.9) between either untreated cells and digitonin semi-permeabilized cells, or untreated cells and ciliobrevin D treated cells, as expected. For velocity components along the cilia axis, there was no significant difference between untreated cells and ciliobrevin D treated cells (p>0.05). Meanwhile, there was a significant difference between velocity distributions in untreated and digitonin semi-permeabilized cells for both the anterograde velocities (p=0.0004) and retrograde velocities (p=0.003) along the cilia axis.

These differences between distributions were further analyzed by performing an effective histogram subtraction. The anterograde and retrograde velocities were separately binned to create histograms with an optimum bin size designed to generate an unbiased estimate of the distributions (*Scott, 1979*). Using nonlinear least-squares, the live cell velocity histogram was fitted to a mixed distribution consisting of fraction f of a Gaussian with unknown mean and variance and fraction (1−f) of the permeabilized cell distribution. For anterograde velocities, the fitted fraction in the additional Gaussian was f = 0.22 ± 0.13. The fitted Gaussian had a mean at 0.41 ± 0.14 µm/s and a standard deviation 0.26 ± 0.12 µm/s. An F-test was used to establish that the fitted mixed distribution fitted the live cell anterograde velocity data significantly better compared to the permeabilized cell distribution alone (p=$10^{-5}$, justifying the three additional degrees of freedom). For retrograde velocities, we fitted a fraction f = 0.24 ± 0.06 in the additional Gaussian, which had a mean at 0.30 ± 0.10 µm/s and a standard deviation 0.28 ± 0.06 µm/s. The F-test statistic established that the mixed distribution yielded a significant improvement in fitting the retrograde velocity data (p=$10^{-5}$).

## FRAP data analysis

The measured fractional fluorescence intensity in a narrow segment of the cilium as a function of time after photobleaching was fitted to a model of one-dimensional diffusion in a bounded region. The cumulative density of bleached molecules between positions 0 and x at time t is given by,

$$H(x,t) = \frac{xa}{L} + \sum_{n=0}^{\infty} \left[ \frac{2L}{\pi^2 n^2} \sin\left(\frac{n\pi a}{L}\right) \sin\left(\frac{n\pi x}{L}\right) \exp\left(\frac{-\pi^2 D n^2 t}{L^2}\right) \right],$$

where the region from 0 to a is bleached, L is the length of the cilium, and D is the diffusion coefficient.

If $f_b$ is the fraction of molecules bleached within the bleached region, and $f_m$ is the fraction of molecules that are mobile, then the fractional intensity measured between positions $a_1$ and $a_2$ at time t after photobleached is given by

$$f = 1 - (1 - f_m) f_b - \frac{f_b f_m}{1 - f_b a/L} \frac{[H(a_2, t) - H(a_1, t)]}{a_2 - a_1}$$

This functional form is used in a least-squares fit for the diffusion coefficient D.

The fraction bleached is estimated as

$$f_b = 1 - f(0)/I_0,$$

where f(0) is the measured fractional intensity in the bleached region at time 0, and $I_0$ is the intensity in the unbleached region at time 0.

The fraction mobile is estimated as

$$f_m = \frac{f_{max}(1 - f_b a/L) - (1 - f_b)}{f_b(1 - a/L)},$$

where $f_{max}$ is the maximal fractional intensity at long times.

## Acknowledgements

We thank Qicong Hu for initiating the project, Ray Goldstein for stimulating discussions and Alice Ting, Kirk Mykytyn, James Chen, and Jennifer Lippincott-Schwartz for reagents.

# Additional information

### Competing interests

WJN: Reviewing editor, *eLife*. The other authors declare that no competing interests exist.

### Funding

| Funder | Grant reference number | Author |
| --- | --- | --- |
| National Science Foundation | EFRI-MEK | W James Nelson |
| National Institutes of Health | GM35527 | W James Nelson |
| National Institutes of Health | GM089933 | Maxence V Nachury |
| March of Dimes | 5-FY09-112 | Maxence V Nachury |
| Damon Runyon Fellowship | DRG 2087-11 | David K Breslow |

The funders had no role in study design, data collection and interpretation, or the decision to submit the work for publication.

### Author contributions

FY, DKB, Conception and design, Acquisition of data, Analysis and interpretation of data, Drafting or revising the article; EFK, WJN, MVN, Conception and design, Analysis and interpretation of data, Drafting or revising the article; AJS, Conception and design, Analysis and interpretation of data

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
