## [Decision Letter]

[Editors’ note: this article was originally rejected after discussions between the reviewers, but the authors were invited to resubmit after an appeal against the decision.]

Thank you for choosing to send your work entitled “Single Molecule Imaging Reveals a Major Role for Diffusion in the Exploration of Ciliary Space by Signaling Receptors” for consideration at *eLife*. Your article has now been peer reviewed and we regret to inform you that your work will not be considered further for publication. Your submission has been evaluated by 3 reviewers, one of whom, Vivek Malhotra, is a member of our Board of Reviewing Editors, and the decision has been discussed further with one of *eLife’s* Senior editors.

Based on extensive discussions between the reviewers before we reached this decision, unfortunately the consensus is that we cannot accept your manuscript for publication by *eLife*. In a previous publication you have already reported the diffusion of proteins in the ciliary membrane (Science 2010). While we agree that in your current paper you have sought to search for the mechanism, the data provided is not convincing. For example, the use of digitonin to test the significance of ATP in the migration of SSTR3 is less than satisfactory. It is not clear if this treatment is sufficient to deplete the ciliary (cellular) ATP levels.

An even bigger concern is the use of ciliobrevin as a reagent to test the potential involvement of Dynein. Is the function of Dynein completely inhibited under your experimental conditions? Your findings that this compound might also be inhibiting Kinesin cast a doubt on the efficacy of ciliobrevin. The other major concern is whether the tagged SSTR3 is correctly targeted and functional? Addressing the functional significance of the tagged SSTR3 is an important issue, albeit technically difficult. However, given the other concerns, the consensus is that your manuscript is currently not suitable for further consideration at *eLife*.

We are therefore, with regret, returning your manuscript to you with the hope that you will find our comments helpful in revising the manuscript for submission elsewhere. The original reviews, on which the discussions between the reviewers were based, are below for reference.

**Reviewer #1:**

This is an interesting and surprising finding and should be published. I would, however, like to raise the following concerns.

Does digitonin permeabilization really affect intracellular ATP levels? It would be better to use the more standard/reliable approach of treating cells with sodium azide and deoxglucose.

The data on the use of ciliobrevin to inhibit dynein is alright but is there a more specific procedure to address the involvement of dynein?

Minor comment:

In the first few lines of the Results and Discussion the authors refer to SST3 as a GPRC. It should be GPCR instead.

**Reviewer #2:**

I was not very excited by the work. The insight provided is not great. The authors suggest that diffusion is enough to account for the spread of the receptor. However, receptors are likely concentrated in the Cilia, and diffusion alone cannot account for that. Also, any clustering of components changes the diffusion properties. Further, there are no functional studies. Just because the receptor they looked at can spread out and looks more or less 'o.k.' as far as distribution does not mean that overall function does not depend on active transport. Further, are the transient interactions with motors that they suggest relevant/important for function?

Critical controls are missing. The authors imaged singly biotinylated Somatostatin Receptor 3 (SSTR3, a ciliary GPCR) labelled with fluorescent monovalent streptavidin. We don’t know what the labeling does to the function/transport of the receptor. If the receptor usually forms a complex, but said complex is altered due to the labeling, the ability of the receptor to diffuse would be significantly altered-size and variety of interactions can influence efficacy of diffusion a lot. So, just because a small uncomplexed receptor can diffuse does not mean that a larger one can/will. At the very least, things will depend on size of things. This is ignored.

Overall, this manuscript seems to me to provide an intriguing hint that diffusion may play an interesting role in the process, but it falls far short of providing a definitive understanding of the process, and as such, seems inappropriate to me for *eLife*.

**Reviewer #3:**

Nachury and colleagues address the relative importance of active transport and diffusion in the localization of ciliary membrane proteins. In theory diffusion should account for the exploration of the entire ciliary membrane surface by membrane proteins. A series of experiments using FRAP to measure behaviour of SSTR3 (a membrane protein) are most consistent with diffusion. Furthermore, conditions that reduce active transport such as ATP depletion or inhibition of dynein using ciliobrevin have little effect on the behavior of SSTR3. This would not exclude a role of active transport in restricting membrane proteins to subdomains of the membrane, although this would not appear to be relevant for SSTR3.

This is a technically well-executed study with clear logic behind all the major experiments. It is of importance for understanding cilium function, and therefore suited to publication in *eLife*.

The limitations are that:

1) a single membrane protein is used for the experiments.

2) motor protein inhibition is achieved by a drug ciliobrevin rather than specific targeting of motor complex subunits.

Ideally some data on a second membrane protein would be useful. Addressing point 2 is a little more complex since targeting motor proteins may compromise cilium formation.

[Editors’ note: guidance for resubmission follows.]

Thank you for choosing to send your work entitled “Single Molecule Imaging Reveals a Major Role for Diffusion in the Exploration of Ciliary Space by Signaling Receptors” for consideration at *eLife*. We have discussed your letter of appeal regarding your full submission and we are prepared to consider a revised submission, with no guarantees of acceptance.

It is important that you include the following data:

1) Experiments strengthening the depletion of ATP.

2) On the migration of smoothened by single molecule analysis.

3) The functionality of tagged SSTR3-GFP studied in your experiments.

4) Discuss your previous paper with reference to the new findings.

---

## [Author Response]

*1) ATP depletion inhibits IFT. Does digitonin permeabilization really affect intracellular ATP levels? It would be better to use the more standard/reliable approach of treating cells with sodium azide and deoxglucose [Reviewer 1]. The use of digitonin to test the significance of ATP in the migration of SSTR3 is less than satisfactory. It is not clear if this treatment is sufficient to deplete the ciliary (cellular) ATP levels [editorial letter]*.

We have conducted experiments using cells treated with deoxyglucose and antimycin A and find that the movement of IFT foci is inhibited while SSTR3 movements are globally unaffected. A detailed analysis is presented in Figure 2 and shows that – like digitonin – deoxyglucose/antimycin treatment inhibits active transport of single SSTR3 molecules. We have also measured ATP levels after antimycin/deoxyglucose treatment (Figure 2) and digitonin permeabilization (Figure 3), and find that digitonin is slightly more effective than antimycin/deoxyglucose in reducing ATP levels, but both treatments remove more than 90% of cellular ATP.

*2) Inhibition of IFT motor proteins. The data on the use of ciliobrevin to inhibit dynein is alright but is there a more specific procedure to address the involvement of dynein? [Reviewer 1]; motor protein inhibition is achieved by a drug ciliobrevin rather than specific targeting of motor complex subunits [Reviewer 2]*.

Ciliobrevin is currently the only published small molecule, membrane permeable inhibitor of dynein that can be added to live cells. In published experiments by Firestone et al., ciliobrevin neither inhibited the activity of kinesin-1 in a microtuble gliding assays, nor the ATPase activity of kinesin-1 or kinesin-5 ([9], Nature *484:125-9).* The only alternative treatment that we could use to inhibit IFT motors would be to deplete dynein 1b or kinesin-II using siRNA. However, IFT motors are required for ciliogenesis, and siRNA treatments will lead to morphologically aberrant cilia in which indirect effects on IFT rates and membrane diffusion will make data impossible to interpret.

*An even bigger concern is the use of ciliobrevin as a reagent to test the potential involvement of Dynein. Is the function of Dynein completely inhibited under your experimental conditions [editorial letter]*?

We treat cells with 50 μM ciliobrevin D, a concentration shown by Firestone et al. to completely inhibiting dynein 1 in cells and *in vitro.* Beyond the concern of remaining dynein 1b activity, we observed a complete block of IFT foci movement while active transport of single SSTR3 molecules appeared uninhibited. To this day, more that 50 published studies have measured IFT foci movements and models have been generated based on the assumption that cargo movements were directly correlated to the movement of these IFT foci. Even if a small percentage of dynein 1b remained active in 50 μM ciliobrevin D, our results suggest – for the first time – that following IFT foci movements cannot be equated to cargo movements.

*3) A single membrane protein is used for the experiments*.

We conducted a series of experiments following single molecules of the Hedgehog signaling intermediate Smoothened (Smo) in cilia. We find that Smo – like SSTR3 – moves in cilia by a combination of diffusion and active transport (Figure 3). The proportion of time spent undergoing active transport are similar for Smo and SSTR3.

*4) Functionality of AP-SSTR3-GFP. Critical controls are missing. The authors imaged singly biotinylated Somatostatin Receptor 3 (SSTR3, a ciliary GPCR) labeled with fluorescent monovalent streptavidin. We don't know what the labeling does to the function/transport of the receptor [Reviewer 2]. The other major concern is whether the tagged SSTR3 is correctly targeted and functional? Addressing the functional significance of the tagged SSTR3 is an important issue, albeit technically difficult [editorial letter]*.

We present data in Figure 2 showing that AP-SSTR3-GFP undergoes agonist-induced endocytosis from the plasma membrane, a key signaling event of functional GPCRs, indicating that the fusion protein is indeed functional. AP-SSTR3-GFP is a targeted to the cilia, as expected.

*If the receptor usually forms a complex, but said complex is altered due to the labeling, the ability of the receptor to diffuse would be significantly altered-size and variety of interactions can influence efficacy of diffusion a lot [Reviewer 2]*.

We addressed the possibility that binding of monovalent streptavidin (mSA) affects the mobility of biotinylated AP-SSTR3-GFP by comparing the diffusion of mSA-labeled AP-SSTR3-GFP to the diffusion of unlabeled protein by FRAP (Figure 2). The absence of a significance difference in the apparent diffusion coefficients suggests that our labeling strategy does not alter the behavior of single molecules.

*5) Other points. In a previous publication you have already reported the diffusion of proteins in the ciliary membrane (Science 2010) [editorial letter]*.

Our Science paper provided evidence for a diffusion barrier at the base of the cilium between the ciliary membrane and apical plasma membrane. We did not conduct any experiments that analyzed whether ciliary protein movements *within* cilia were the result of passive diffusion or active transport. We have clarified the presentation of our past results in the text.

*Further, are the transient interactions with motors that they suggest relevant/important for function [Reviewer 2]*?

The regulation of signaling function by IFT is an exciting hypothesis that is compatible with our observations and this is discussed in the final paragraph of the Discussion.